# Antibacterial Activity of *Thymus vulgaris* L. Essential Oil Vapours and Their GC/MS Analysis Using Solid-Phase Microextraction and Syringe Headspace Sampling Techniques

**DOI:** 10.3390/molecules26216553

**Published:** 2021-10-29

**Authors:** Julien Antih, Marketa Houdkova, Klara Urbanova, Ladislav Kokoska

**Affiliations:** 1Department of Crop Sciences and Agroforestry, Faculty of Tropical Agrisciences, Czech University of Life Sciences Prague, Kamycka 129, 165 21 Prague-Suchdol, Czech Republic; antih@ftz.czu.cz (J.A.); houdkovam@ftz.czu.cz (M.H.); 2Department of Sustainable Technologies, Faculty of Tropical Agrisciences, Czech University of Life Sciences Prague, Kamycka 129, 165 21 Prague-Suchdol, Czech Republic; urbanovak@ftz.czu.cz

**Keywords:** antimicrobial activity, broth microdilution, headspace analysis, respiratory infections, thyme, vapour phase

## Abstract

While the inhalation of *Thymus vulgaris* L. essential oil (EO) is commonly approved for the treatment of mild respiratory infections, there is still a lack of data regarding the antimicrobial activity and chemical composition of its vapours. The antibacterial activity of the three *T. vulgaris* EOs against respiratory pathogens, including *Haemophilus influenzae, Staphylococcus aureus*, and *Streptococcus pyogenes,* was assessed in both liquid and vapour phases using the broth microdilution volatilisation (BMV) method. With the aim of optimising a protocol for the characterisation of EO vapours, their chemical profiles were determined using two headspace sampling techniques coupled with GC/MS: solid-phase microextraction (HS-SPME) and syringe headspace sampling technique (HS-GTS). All EO sample vapours exhibited antibacterial activity with minimum inhibitory concentrations (MIC) ranging from 512 to 1024 μg/mL. According to the sampling technique used, results showed a different distribution of volatile compounds. Notably, thymol was found in lower amounts in the headspace—peak percentage areas below 5.27% (HS-SPME) and 0.60% (HS-GTS)—than in EOs (max. 48.65%), suggesting that its antimicrobial effect is higher in vapour. Furthermore, both headspace sampling techniques were proved to be complementary for the analysis of EO vapours, whereas HS-SPME yielded more accurate qualitative results and HS-GTS proved a better technique for quantitative analysis.

## 1. Introduction

Pneumonia is an acute lower respiratory infection (ALRI) mainly caused by bacterial pathogens, including Gram-positive bacteria such as *Staphylococcus aureus, Streptococcus pneumoniae*, and *Streptococcus pyogenes,* or Gram-negative strains such as *Haemophilus influenzae* [1,2,3]. For decades, they have been amongst the major causes of morbidity and mortality worldwide; this is particularly significant in low-income countries where vulnerable populations such as children under five years old, the elderly, and the immunocompromised population are at higher risk [4]. In 2019, ALRIs remained the world’s most deadly communicable diseases with nearly 2.6 million deaths and accounted for 15% of deaths in children under 5 years of age [5,6,7]. To reduce the fatality of these infections, the recommended treatment is the administration of a full course of appropriate antibiotics. Currently, nebulised antibiotics offer significant advantages over intravenous and oral therapies; amongst others, inhalation therapy allows to deliver high drug concentrations and to have more efficient and faster action while reducing the risk of side effects compared to systemic administrations [8]. Today, such treatments are already included in care protocols, especially for cystic fibrosis. For example, nebulised tobramycin, colistin, and aztreonam lysine are commonly used in Europe as inhaled antibiotics against chronic *Pseudomonas aeruginosa* infections [9]. However, difficulties with implementing optimal nebulisation techniques and the lack of robust clinical data have limited the widespread adoption of aerosol treatments [10,11,12]. Among other reasons, this can be attributed to both particle-related and patient-related factors. For instance, efficient delivery of aerosolised antibiotic particles in the lower parts of the lungs rests on the aerodynamic behaviour of the particles: larger particles preferably accumulate in the oropharyngeal area, while smaller particles deposit in the lower airways. Consequently, this alters drug delivery in the lungs as small particles carry fewer active substances [13]. Furthermore, the distribution of nebulised particles depends on the breathing manoeuvre used by the patient: rapid and forceful inhalations will see an increased deposition of drug in the upper airways, whereas slow and deep inhalations deposit particles in the lower part of the respiratory system [14]. As a result, drug delivery is also affected by the quality of inhalations which hinge on the patient’s age (especially children and elderly), the severity of the respiratory disease, or even the physical capability to perform a breathing manoeuvre [15].

In this context, plant-derived preparations, and more specifically, essential oils (EOs), are interesting alternatives in the development of novel antimicrobial volatile agents [16,17]. EOs’ typical physicochemical feature is high volatility at room temperature. Therefore, compared to aerosolised antibiotics, EO vapours are easy to inhale without requiring any specific breathing technique. At the same time, this characteristic allows for a uniform distribution of EOs’ active substances, at a significant concentration, in the lower section of the lungs [18]. As an example, a recent clinical trial study has already demonstrated that a *Thymus vulgaris* L. EO inhalation therapy would improve the respiratory conditions in patients with ventilator-associated pneumonia. Among other benefits, this treatment would facilitate the clearance of the mucous membrane, improve gas exchange in alveoli, and thus increase the oxygen saturation in patients’ blood [19]. Furthermore, EOs are complex mixtures containing a broad spectrum of chemically diverse compounds that work in synergy to provide their antibacterial properties. Their composition includes hydrocarbons (terpenes and terpenoids) and oxygenated compounds (phenols, aldehydes, alcohols, ketones, esters, and lactones) as well as sulphur or nitrogen substances such as isothiocyanates and pyrazines synthesised by certain plant families [20,21,22]. Because of their well-studied chemical compositions, EOs can be used for medicinal purposes if their long-standing usage and experience can be demonstrated [23]. Amongst the nearly 30 EOs approved by the Committee on Herbal Medicinal Products (HMPC) of the European Medicines Agency (EMA), the essential oil of *T. vulgaris* is one of the most important plant species used traditionally to treat respiratory disorders, including cough, bronchitis, laryngitis, and other respiratory congestions [24]. Moreover, several reports are validating the in vitro antibacterial activities of this EO on respiratory pathogens such as *S. aureus* and *P. aeruginosa*. Today, its composition is well known and comprises components such as thymol methyl ether, *p*-cymene, 1,8 cineol, and α-pinene, but it is widely believed that its antimicrobial activity is related to its content in two phenolic monoterpenes: thymol and carvacrol [25,26,27]. *T. vulgaris* is also used in complex herbal formulations. For instance, Bronchipret (Bionorica, Neumarkt, Germany) is an over-the-counter oral treatment consisting mainly of *Hedera helix* L. and *T. vulgaris* extracts. This herbal product is recommended in acute inflammation of the respiratory tract associated with cough and thick mucus [7]. Although recent studies suggest that the EO of *T. vulgaris* produce better results in antimicrobial activities in the vapour phase than their liquid phase [17,28,29], there is still a lack of available data regarding its growth-inhibitory effect against pneumonia-causing bacteria [28].

As opposed to well-established methods for the testing of antimicrobial susceptibility in both solid (i.e., agar disc diffusion) and liquid (broth microdilution) media, there is no standardised antimicrobial assay to study EO vapour phases according to, for example, the Clinical and Laboratory Standards Institute (CLSI) [30,31,32]. Today, disc volatilisation is the most common method to examine the in vitro growth inhibition of EOs’ volatile agents [33]; however, similar methods have already been developed and used (e.g., dressing model volatilisation test [34], or airtight apparatus disc volatilisation methods [28]). Unfortunately, such tests have limitations: they are generally not designed for high-throughput screening, some only allow the evaluation of a single concentration of each sample, and usually only provide qualitative data [35]. Furthermore, the lack of standardisation of methods has led to difficulties not only in interpreting but also in comparing antimicrobial growth results. For instance, from all studies examining *T. vulgaris* EO vapour antimicrobial activity, it is not unusual to observe results expressed in different ways, including unit volume of air [28] or even various definitions of minimum inhibitory concentrations (MICs) [36,37].

The antimicrobial properties of EO vapours are determined by the relative volatility and antimicrobial effects of their compounds [38]. It is therefore essential to understand their chemical composition, something that has not been well-explored yet [39]. Nowadays, to examine the content of EOs vapour, static headspace extraction coupled with gas chromatography–mass spectrometry (SHE-GC/MS) offers more advantages compared to traditional liquid extraction of samples. It is a simple, rapid, and solventless technique that requires a small amount of sample and allows the analysis of highly volatile compounds [40]. Consequently, studies examining the antimicrobial activity of EO vapours are also focused on their chemical composition using SHE-GC/MS. As an example, several studies have explored the chemical composition of *T. vulgaris*’s volatile agents using different SHE techniques, including solid-phase microextraction (HS-SPME) [37,41,42], which is currently the preferred technique to examine complex, volatile mixtures in laboratories [43]. This method uses a fused silica fibre coated with a stationary phase that is directly exposed to the sample headspace. After absorption of the analytes to the fibre coating, the sample is transferred into a GC injection port for thermal desorption [40]. HS-SPME sampling is not only a sensitive technique due to the concentration achieved by the fibre but also a selective one thanks to the different coating material available; repeatability is also one of its assets when used with a dedicated autosampler [40,44,45]. However, using fibre coating suffers limitations; for example, it is not uniformly sensitive; therefore, competitive adsorption between volatile agents for the limited number of active sites can be observed. Similarly, selectivity will be different depending on the coating polymer used [46,47]. As a result, GC peak areas might not reflect the exact compounds’; composition and proportion in the headspace. That is why other approaches can be considered for analysing the EO vapour profile, including gas tight syringe headspace sampling (HS-GTS). This technique is the most convenient and inexpensive way to sample highly volatile compounds from the headspace of a closed vessel [40,48]. The gas syringe with a pressure-lock valve is inserted into the headspace, and a fraction of its volume is removed. The gas sample, locked into the syringe, is then transferred and injected into the GC inlet. Despite the above-mentioned advantages, the potential of this method for headspace sampling of EO vapours has not fully been exploited yet. Intending to identify EO vapours with the potential to inhibit the growth of bacteria causing pneumonia, we performed a series of preliminary experiments with herbal products approved by the HMPC of the EMA for the treatment of infectious cough and cold [23]. It encompassed the assessment of the in vitro growth-inhibitory effect of *Pimpinella anisum* L. seed, *Eucalyptus globulus* Labill. leaf, *Thymus vulgaris* L. aerial part, *Mentha × piperita* L. leaf, and *Foeniculum vulgare* Mill. seed EOs using the broth microdilution volatilisation (BMV) method against *H. influenzae*, *S. aureus*, and *S. pyogenes*. This novel method was recently developed by our team [35]; it is a simple, rapid, and simultaneous technique that allows the assessment of EOs antibacterial activities at different concentrations in both liquid and vapour phases. As a result of these exploratory tests, *T. vulgaris* EOs was selected for further evaluation due to the lowest MICs it generated (J. Antih, M. Houdkova, and L. Kokoska, unpublished data). We therefore investigated the in vitro growth-inhibitory effect of *T. vulgaris* EOs in both liquid and vapour phases against the above-mentioned respiratory pathogens and compared their chemical compositions using GC-MS. Then, with the aim of optimising a protocol for the chemical characterisation of EO vapours, we performed a time series of headspace analyses comparing both HS-SPME and HS-GTS sampling techniques.

## 2. Results

### 2.1. Antimicrobial Activity

In this study, essential oil samples from dry plant material of *T. vulgaris* from three different suppliers were tested against three standard bacterial strains associated with respiratory infections using the BMV method (Table 1). All EOs presented a certain degree of antibacterial efficacy ranging from 512 to 1024 μg/mL in both liquid and vapour phases. Supplier C’s EO was the most active, with the lowest MICs value of 512 μg/mL in both liquid and vapour phases for the three bacteria strains. Similarly, *S. pyogenes* and *H. influenzae* growth were more affected by the EO of supplier B than *S. aureus* with MICs at 512 μg/mL and 1024 μg/mL in both broth and agar, respectively. On the contrary, the least effective EO source was from supplier A: results showed mild efficacy against *H. influenzae* (512 μg/mL), whereas a weaker inhibitory effect of 1024 μg/mL was observed against both *S. aureus* and *S. pyogenes*. Likewise, each EO affected the growth of *S. pyogenes*, *H. influenzae*, and *S. aureus* similarly in both liquid and vapour phases. No discrepancy between broth and agar results was observed on the tested strains. *H. influenzae* was the most susceptible bacterial strain (MICs = 512 μg/mL for all EOs tested) followed by *S. pyogenes* (MICs = 512 μg/mL supplier B and C and MIC = 1024 μg/mL for supplier C), while *S. aureus* was the least sensitive (MIC = 512 μg/mL supplier C and MICs = 1024 μg/mL for supplier A and B).

### 2.2. Chemical Analysis of EOs

In this investigation, *T. vulgaris* EOs from three different suppliers (A, B, and C) were extracted with respective yield values of 0.73, 1.23 and 1.25%. All EOs presented a *strong* herbaceous *fragrance* while being of different shades of orange colour. The complete chemical analyses of all samples are provided in Table 2 as well as in Figure 1a,b. Using the HP-5MS column, 54 compounds were identified in EOs of suppliers A and B, whereas 62 components were found in supplier C’s EO, representing 99.55, 99.65, and 99.62% of their respective total constituents, respectively. Similarly, using the DB-HeavyWAX column, 44, 43, and 36 components were determined, which constituted 99.30, 99.68, and 99.61% of the volatile oils, respectively. In the three samples analysed, monoterpenoids represented by thymol (phenolic monoterpene) followed by sesquiterpenoids mainly represented by β-caryophyllene were the two dominant groups of volatile agents identified. To a lesser extent, other groups were identified, such as oxygenated phenylpropanoids, oxygenated aliphatics, furanoids, and oxygenated diterpenes.

In all *T. vulgaris* EOs, thymol was the most abundant compound. When considering both HP-5MS/DB-HeavyWAX columns, supplier C’s EO percentage values were 48.09/48.65% in contrast to suppliers A and B results—i.e., 38.42/29.52% and 38.93/41.31%, respectively. Likewise, *p*-cymene was the second most abundant constituent in the three EO samples. While in sample C, the peak percentage area represented 12.75/12.90%, its value was twice as high in both supplier A and B EOs—22.15/16.43% and 22.03/25.35%. Carvacrol was the third abounding compound in A, B, and C samples, with percentage values of 10.61/3.84%, 10.85/5.59%, and 10.92/3.60%, respectively. Although the content of γ-terpinene was relatively high in the three samples, the amount detected in EOs from suppliers A and B (2.82/2.02% and 2.78/2.90%) were significantly lower than in supplier C EO (5.22/5.40%). Eventually, in EO samples A, B, and C, other compounds were detected in amounts lower than 1.80/1.08%, 1.82/1.77%, and 1.53/1.38%, respectively. Furthermore, ten compounds, including 4-carene, 4-pentenyl butyrate, α-cubebene, isobornyl acetate, calamenene, isoborneol, aromandendrene, lavandulyl butyrate, *p*-cymen-7-ol, and phytol, were only detected by the DB-HeavyWAX column at an amount lower than 0.86% in supplier A, 1.17% in supplier B, and 1.05% in supplier C.

### 2.3. Chemical Analysis of EOs’ Vapour Phase

In this study, the sampling of the headspace above a mixture of *T. vulgaris* EO and MH broth has been carried out using two different methods of extraction, i.e., HS-SPME and HS-GTS extractions. Headspace chemical compositions were measured every 3 h during a 12-h experiment using the HP-5MS column. Complete analyses are provided in Table 3 and Table 4, as well as Figure 1c,d. Using HS-SPME extraction, a total of 40, 38, and 43 volatile compounds were identified in the samples of suppliers A, B, and C, respectively. This represented 99.85, 99.89, and 99.63% of their respective total constituents at 12 hrs. In contrast, a significantly lower number of compounds was detected when using the HS-GTS extraction method. While 32 constituents were found in EO sample C, only 26 components in samples A and B were found, which accounted for 99.87, 99.95, and 99.71% of their total contents at time 12 h. Regardless of the extraction method used and the three samples analysed, monoterpenoids followed by sesquiterpene hydrocarbons were by far the two most predominant chemical groups of volatile compounds identified in the headspace. Other chemical categories present in minor amounts (oxygenated phenylpropanoids, oxygenated aliphatics, and furanoids) were identified using HS-SPME extraction only. On the other hand, cyclic ethers such as furan derivatives were only present in samples obtained by HS-GTS extraction.

Using the HS-SPME extraction method, the most abundant volatile substance across all *T. vulgaris* was *p*-cymene. Its percentage values during the whole experiment were rather similar in the headspace of samples A, B, and C, ranging from 54.57 to 58.61%, 69.91 to 74.50%, and 58.38 to 67.21%, respectively. Likewise, the second most abundant compound was γ-terpinene. While in sample A the peak percentage area was between 19.74 and 20.45% during the 12 h experiment, these values were lower in both supplier C and B, ranging from 12.37 to 16.18% and 7.07 to 9.30%, respectively. Thymol was the third abounding compound in A, B, and C samples during the entire 12-h period, with percentage values ranging from 2.61 to 3.36%, 2.13 to 5.25%, and 3.72 to 5.27%, respectively. Similarly using HS-GTS extraction, *p*-cymene (47.05 to 50.73% for sample A, 52.28 to 57.41% for sample B, and 44.80 to 49.28% for sample C), as well as γ-terpinene (13.40 to 17.45%, 5.76 to 6.99%, and 9.54 to 11.85% for sample A, B, and C, respectively), were the first two most abundant compounds in the headspace for all EO samples and during the full experiment. However, the third volatile substance detected in a significant amount was α-pinene, with percentage values over the 12-h experiment between 5.74 and 6.83%, 6.34 and 8.98%, and 6.90 and 7.96% for EO samples A, B, and C, respectively. Other differences have been observed when comparing the two sampling methods. Firstly, the number of compounds detected using HS-SPME was higher than when using HS-GTS extraction—i.e., on average, 41 versus 28 compounds. Then, the chemical analysis showed that when using HS-SPME extraction, a larger number of sesquiterpene hydrocarbons (14 components) were found in amounts lower than 3.09%, 1.59%, and 2.94% in EO samples A, B, and C, respectively. In contrast, using the HS-GTS method, only six compounds were found, including α-copaene, β-bourbonene, β-caryophyllene, γ-muurolene, γ-cadinene, and δ-cadinene, at amounts lower than 0.38% in sample A, 0.28% in sample B, and 0.58% in sample C. Similarly, percentage values of phenolic monoterpenes and derivatives were considerably higher using HS-SPME sampling method (overtime for samples A, B, and C, lower than 3.14%, 5.25%, and 5.27%, respectively) than HS-GTS extraction (values lower than 0.67%, 0.49%, and 0.58%, respectively). Eventually, despite the above-mentioned discrepancies, the headspace analysis of both sampling methods showed that there were no significant changes in the chemical composition in the vapour of the three EO samples of *T. vulgaris* over time.

## 3. Discussion

As reported by Houdkova and Kokoska [52], several assays have previously been developed for the evaluation of the antibacterial activity of volatile plant compounds in the vapour phase. However, there is still a lack of standardised methods, something that makes any interpretation and comparison difficult [52]. For instance, three different tests have been identified to investigate *T. vulgaris* EO vapours activity, and results are described differently according to each author. A study led by Inouye et al. [28] assessed the growth-inhibitory effects of two *T. vulgaris* EO vapours using the airtight box disc volatilisation method. Introducing the minimum inhibitory dose (MID) expressed in mg/L air, they determined that the vapour of carvacrol chemotype EOs were more active against Gram-negative *H. influenzae* than against Gram-positive strains such as *S. pyogenes* and *S. aureus* (MIDs of 3.13, 6.25, and 12.5 mg/L air, respectively). In this study, we reported similar results: *H. influenzae* was more susceptible—i.e., MIC = 512 µg/mL or 128 µg/cm^3^ considering the volume of the entire well—to the vapour of our three thymol chemotype EOs than the *S. pyogenes*, and *S. aureus* (MICs comprised in between 1024–512 µg/mL or 256–128 µg/cm^3^ for both bacteria strains). It is largely admitted that Gram-negative bacteria are more resistant to EOs than Gram-positive ones [53,54]. This weak antibacterial activity was attributed to the presence of hydrophilic polysaccharide chains in the outer membrane structure, preventing hydrophobic EOs from reaching the bacteria cell membrane [55]. One reason that could explain the higher susceptibility of *H. influenzae* to EOs would be the more hydrophobic nature of its outer membrane composed of oligosaccharide shorter chains [28]. This was confirmed by Reyes-Jurado et al. [56], who demonstrated that compounds including *p*-cymene, linalool, and thymol were able to disintegrate such outer membrane structures. Furthermore, various studies have subsequently assessed the growth-inhibitory effects of *T. vulgaris* EO vapour using vapour diffusion assay developed by Lopez et al. [37]. Nedorostova et al. [36] reported a MIC of 17 μL/L against *S. aureus,* which is, according to the author, equivalent to the result of Inouye et al. [28] against the same bacterium (MID = 12.5 mg/L of air). Similarly, Kloucek et al. [57] have observed that the vapour of *T. vulgaris* EO consisting mainly of geraniol had a MIC = 125 μL/L against *S. aureus* as well. In contrast to these findings, MIC values recorded in our study were usually higher—i.e., 1024–512 µg/mL (or 256–128 µg/cm^3^ considering the volume of the entire well). The discrepancy in results could be firstly explained by the quality of the EO samples used and their chemical compositions [28], but also by the disparate strains of bacteria used in the different antimicrobial assays [58]. Most importantly, this variation may also be attributed to the diverse methods used to explore the antimicrobial effect of *T. vulgaris* EO vapour, allowing various interpretations of the MIC [35,37,57]. That is why the BMV assay presented here is a powerful alternative to the previously developed techniques. While their designs only enable the assessment of EO vapour at a single concentration [57], the BMV assay is conceived to evaluate the EOs’ in vitro growth-inhibitory effect in liquid and gaseous phases simultaneously and at different concentrations, something that allows fast comparison of MIC values in both liquid and solid media. Based on broth microdilution [59] and disc volatilisation (DV) assays [33], the BMV experiments are conducted on standard 96-well immune plates, which offer several advantages such as cost and labour efficiency [57,60]: microplates are standard laboratory equipment that are commonly available and compared, for example, to the special airtight experimental apparatus used in certain methods such as in Seo et al. [61]. Furthermore, microplates can also be employed in fully automated workstations, unlike Petri dishes used in DV assays which therefore suffer from a lack of repeatability [62]. Other studies developed assays using microplates for detecting volatile substances antimicrobial activity, such as the vapour-phase-mediated patch assay of Feyaerts et al. [63]. However, contrary to the BMV assay, their designs only allow to determine relative microbial inhibition values, and their main limitation lies in only providing qualitative results [64]. Eventually, another asset of our assay is that modifications can be easily implemented for new applications. For instance, Netopilova et al. [17] modified the test to explore the combinatory effects of volatile substances using a chequerboard design and thus determine fractional inhibitory concentration (FIC) indices, something not possible with other methods. Overall, these features allow our method to be suitable for high-throughput screening and thus be a simple, fast, and reliable assay as well as providing reproducible and quantitative results [65]. Nevertheless, despite its numerous advantages, serially produced microplates are not designed for volatile substance testing, something that contributes to weaknesses shown by our assay. For example, only a limited volume of agar can be pipetted into each flange of the lid; this could impact the bacterial growth and thus affect the results. Most importantly, when testing EO vapours, the BMV assay also faces specific problems linked to the physico-chemical nature of EOs and, more particularly, their high volatility, viscosity, and hydrophobicity [52]. Firstly, volatility allows a loss of active substances by evaporation that can happen at different steps of the protocol, including during sample handling and experiment preparation, complications that are shared by all tests assessing EO properties [66,67]. More specifically to volatilisation assays, the level of vapour transition from the matrix into which the EO is included and its distribution into the inner atmosphere of the well during the experiment are two critical factors that may affect the results. For instance, the matrix influences both the intensity and the speed of the evaporation [35]. In our assay, a broth medium was used, which according to Orchard et al.’s [68] observations, would slow the level of vapour transition during the experiment and thus affect the bacterial growth. Similarly, as described by Reyes-Jurado et al. [69], the hydrophobicity and viscosity properties of the EO may also cause its uneven distribution through the broth medium, something that could also alter the distribution of the volatile agents into the well’s atmosphere. That is why the concentrations in the vapour phase used in our experiment should only be considered as indicative values.

The antibacterial properties of *T. vulgaris* are mainly attributed to the chemical composition of its EO, which has already been extensively studied. Its major constituents are mainly monoterpenoids, such as carvacrol, thymol, γ-terpinene, and *p*-cymene, but also sesquiterpenoids such as β-caryophyllene. Within the same species, the variation of proportions of these compounds defines the EO chemotype, which is named after the predominant constituent identified [70]. Our chemical analyses showed that thymol was the most abundant constituent within our samples, followed by *p*-cymene and carvacrol. This characterises our three EOs as thymol chemotypes. These findings are in accordance with several studies previously published. For instance, Schmidt et al. [71], Grosso et al. [72], and Nikolić et al. [25] reported thymol as the major component of their *T. vulgaris* thymol chemotype EOs (peak percentage area of 38.8%, 41.6%, and 49.1%, respectively); *p*-cymene was the second most abundant with 24.0%, 28.9%, and 20.0%, respectively. However, instead of carvacrol, γ-terpinene was detected as the third most abundant constituent (9.5%, 5.1%, and 4.2%, respectively). The variations in yields and concentrations of volatile compounds in our samples could be attributed to several factors occurring at different stages, from the growing conditions of the plants, the harvesting period to the storage conditions of the plant materials by the commercial suppliers [18,73]. For instance, Nezhadali et al. [56] showed that *T. vulgaris* harvested in the same location but at different stages of the plant growth resulted in different yields of EOs: the highest oil yield (1.39%) was obtained during the flowering period, whereas the lowest yield (0.83%) corresponded to the fruiting stage. Similarly, he reported that the concentration of thymol between those periods dropped from 63.01% to 38.23% of the total EO content. Subsequently, the characterisation of the EOs was carried out by GC/MS using two capillary columns of different polarities. The concomitant use of a polar column (DB-HeavyWAX) along with a non-polar (HP-5MS) allows revealing overlapped signal peaks and thus improves the identification of the separated compounds. Hudaib et al. [74], who analysed the chemical profile of *T. vulgaris* EO by GC/MC using the same approach, demonstrated that the polar column helped to enhance the resolution between compounds co-eluting on a non-polar column such as the couples (α-thujene, α-pinene) or (sabinene, β-pinene). This has also been observed in our study with, for instance, the couple (*p*-cymene-limonene). Moreover, as described by Fan et al. [75], while the main constituents of an EO are equally detected by both columns, a fraction is identified by either of them: in our study, 23 compounds out of 75 were detected using HP-5MS only—i.e., representing on average 5.82% of the compounds identified within the three samples—and 10 compounds (1.88% on average) with DB-HeavyWAX. The difference in detection recorded would be the result of the different polarity and material of the columns used [76]. Overall, these results suggest that complementing a non-polar with a polar column provides a more precise picture of the EO analysed than if displayed individually.

The characterisation of *T. vulgaris* EO vapours was carried out using two different sampling methods: HS-SPME and HS-GTS. In recent years, HS-SPME has become the preferred laboratory method for identifying EOs volatile compounds: it is a simple, fast, cost-effective, selective, and sensitive method that provides high-quality results [40,42,60]. With the aim to simulate the conditions of the antimicrobial susceptibility testing experiments performed in this study, the EO samples were prepared identically to the most active EO during the BMV assay (i.e., incubation temperature was at 37 °C and EOs dissolved in Mueller–Hinton (MH) broth medium concentrated at 512 µg/mL). In addition, we used a mixed coating material (DVB/CAR/PDMS) that gives better extraction yields for both polar and non-polar volatile constituents than simple fibre coatings [41,77]. As a result, our investigation revealed that although *p*-cymene and γ-terpinene were abundant in the headspace, the amount of thymol extracted by the coated fibre was unusually low (peak percentage area lower than 5.27% across the three EO samples). This observation is in contradiction with what was described in previously published research. For instance, Lugo-Estrada et al. [41], Soleimani et al. [78], and Nezhadali et al. [79], who investigated *T. vulgaris* EO vapour composition, all reported that thymol was the most abundant constituent of the headspace (peak percentage area of 34.28%, 28.50%, and 45.45%, respectively). As described by Adam et al. [80], efficient extractions of EO volatile compounds depend on optimised experimental parameters such as the selection of the fibre coating material and the incubation temperature of the EO sample. The lack of thymol could, therefore, be explained by several reasons. Firstly, it is the selectivity and sensitivity of the DVB/CAR/PDMS coating. Although it proved to be the most universal assembly for the isolation of compounds with diverse physico-chemical properties [81], Soleimani et al. [77] demonstrated in their comparative study that phenolic monoterpenes and, more particularly, thymol, were better extracted by a simple polydimethylsiloxane (PDMS) fibre than using mixed coating materials. Furthermore, as the transfer rate of volatile agents toward the fibre increases with the incubation temperature of the sample [81], the temperature chosen could have potentially limited the extraction of thymol in our investigation. For example, in their research work, Nezhadali et al. [82] compared the HS-SPME extraction efficiency of *T. vulgaris* EO main volatile agents at 25 °C and 50 °C using a water-based matrix. The result showed that the amount of thymol at higher temperatures is almost twice as high as at lower temperatures (73.09% and 45.45%, respectively). This suggests that different experimental conditions using the HS-SPME technique can yield different distributions of EO volatile compounds, and therefore, the result may not necessarily illustrate the actual chemical composition of the headspace above the EO samples. In contrast, the HS-GTS technique provided a different perspective on the constituency of the headspace, perhaps closer to the real distribution of volatile compounds in the vial at equilibrium [83]. Despite this, the results showed a peak percentage area for thymol lower than 0.60% across the three EO samples—even below the levels observed with the HS-SPME method. Using identical experimental conditions with both methods has demonstrated unusually low concentration levels of thymol. The possible explanation may therefore lie in the matrix in which the EO was inserted. As previously mentioned, the hydrophobic nature of volatile compounds worsens their solubility in water-based media (here the MH broth medium), which may reduce their dilution capability and result in an unequal distribution throughout the medium, something that may also affect their distribution into the well’s atmosphere [69].

The HS-GTS method also yielded interesting results with other components; for instance, the amount of α-pinene obtained with HS-GTS sampling (peak percentage area lower than 7.96% across the three EO samples) was at least three times higher than the amount extracted with HS-SPME (peak percentage area lower than 2.82%). Despite a lack of academic studies examining *T. vulgaris* EO using HS-GTS, these findings are consistent with the data available in other publications. Coleman et al. [84], for instance, compared the distribution of volatile constituents in the headspace above a sample of *Juniperus virginiana* EO using both HS-SPME and HS-GTS techniques. He observed that α-pinene dominated the headspace with 88.0% of the total composition when using HS-GTS, whereas it only exhibited 32.4% with the HS-SPME technique. According to the author, this difference may be explained by the sorption behaviour of α-pinene over time when using a coated fibre assembly. In fact, he demonstrated that the percentage composition of volatile compounds changes significantly with the fibre exposure time, with α-pinene levels dramatically descending with greater time exposure, while high molecular weight components increased with the sampling time.

When comparing the chemical profiles of *T. vulgaris* EO in the liquid phase with the headspace analyses, we observed significant differences between both phases. The EO liquid phase contained a greater number of substances with a variable distribution, whereas the headspace composition detected a smaller number of compounds represented mostly by highly volatile substances. This phenomenon is in accordance with what was already described in previous research [83]. Nevertheless, the three EO samples showed identical concentrations of antimicrobial activities during BMV experiments in both liquid and vapour phases. This raises the question as to the levels of the active concentration of the EO compounds in the vapour, and more particularly thymol as its main antimicrobial constituent [57]. As a phenolic compound, thymol is very stable, moderately soluble in water and of low volatility, and was detected in a lower amount in the headspace—i.e., peak area percentage lower than 5.27% (HS-SPME) and 0.60% (HS-GTS)—than in the sample of EO diluted in the broth (max. 48.65%). By contrast, α-pinene, which is highly volatile and extremely insoluble in water, followed the opposite trend (peak area percentage lower than 2.82% (HS-SPME), 7.84% (HS-GTS), and 0.62% in the sample of EO diluted in the broth). This was observed in previous studies and is explained by the difference in volatility: when the EO is introduced into a closed environment, volatile compounds start to diffuse at different rates according to their molecular weight until they reach equilibrium [69]. Despite its slower diffusion rate, we could possibly argue that the low levels of thymol detected in the headspace was sufficient to generate the same antimicrobial activity as its amount in the EO concentrated at 512 μg/mL. To compare, Wang et al. [84] observed that the active concentration of thymol in its vapour phase samples against oral pathogens was between 100–400 μg/mL. Although the correlation between the concentration of essential oil and thymol were not the subject of this study, the results strongly suggest that the effect of antimicrobial constituents of *T. vulgaris* EO (e.g., of thymol) is higher in vapour than in the liquid phase. To further support this, an additional investigation focused on thymol behaviour in vapour would be needed.

## 4. Materials and Methods

### 4.1. Plant Material and Sample Preparation

Dried aerial part bulk of *Thymus vulgaris* L. were randomly purchased at three local spice stores and e-shops (supplier A = Kralovství chuti s.r.o., Prague, Czech Republic; supplier B = Byliny Mikes s.r.o., Číčenice, Czech Republic; supplier C = Lbros s.r.o., Vrchlabí, Czech Republic). Subsequently, plant materials were ground and homogenised using a Grindomix apparatus (GM 100 Retsch, Haan, Germany). The residual moisture content was evaluated gravimetrically at 130 °C for 1 h by Scaltec SMO 01 Analyzer (Scaltec Instruments, Gottingen, Germany) in triplicates and results were expressed as arithmetic averages (15.29%, 14.54%, and 13.13% for suppliers A, B, and C, respectively).

### 4.2. Hydrodistillation of the Essential Oils

EOs were extracted by hydrodistillation of the ground material following the indication provided by the European Pharmacopoeia [85]: 100 g of ground plant materials placed in 1 L of distilled water was distilled for 3 h using Clevenger-type apparatus (Merci, Brno, Czech Republic). Since hydrodistillation is one of the commonly used methods for commercial production of *Thymus vulgaris* L. EO, the properties of samples prepared in our study should be similar to those commercially available [86]. Eventually, the extracted EOs were stored in sealed glass vials at 4 °C until further handling.

### 4.3. Bacterial Strains and Culture Media

In this study, the following standard strains of the American Type Culture Collection (ATCC) were used: *H. influenzae* ATCC 49247, *S. aureus* ATCC 29213, and *S. pyogenes* ATCC 19615. Both cultivation and assay media (broth/agar) were MH, complemented with Haemophilus Test Medium and defibrinated horse blood for *H. influenzae*, MH only for *S. aureus*, and Brain Heart Infusion when working with *S. pyogenes*. The pH of broths was equilibrated to a final value of 7.6 using Trizma base (Sigma-Aldrich, Prague, Czech Republic). All microbial strains, growth media, and additions were purchased from Oxoid (Basingstoke, Hampshire, UK).

Stock cultures of bacterial strains were cultivated in appropriate media at 37 °C for 24 h prior to the testing, and the bacterial suspension turbidity was adjusted to 0.5 McFarland standard using Densi-La-Meter II (Lachema, Brno, Czech Republic) to reach the final concentration of 10^7^ CFU/mL. Ampicillin, oxacillin, and amoxicillin were purchased from Sigma-Aldrich (Prague, Czech Republic) and assayed as positive antibiotic controls [30].

### 4.4. Antimicrobial Assay

The in vitro growth-inhibitory effect of EOs was assessed using BVM method that allows simultaneous assessment of EOs antibacterial activities at different concentrations in both liquid and vapour phases [4,35]. Experiments were performed using standard 96-well microtiter plates (well volume = 400 μL) covered by tight-fitting lids with flanges designed to reduce evaporation (SPL Life Sciences, Naechon-Myeon, Korea). Initially, 30 μL of agar was pipetted into every flange on the lid, except the outermost flanges, and inoculated with 5 μL of bacterial suspension after solidification of the agar. Subsequently, each EO sample was dissolved in dimethyl sulfoxide (DMSO) (Sigma-Aldrich, Prague, Czech Republic) at a maximum concentration of 1% and diluted in an appropriate broth medium. Seven two-fold serially diluted concentrations of samples starting from 1024 μg/mL were prepared for all EOs. The final volume in each well was 100 μL. The plates were subsequently inoculated with a bacterial suspension using a 96-pin multi-blot replicator (National Institute of Public Health, Prague, Czech Republic). The wells containing inoculated and non-inoculated broth were prepared as growth and purity controls simultaneously. The outermost wells were left empty to prevent the edge effect. Eventually, clamps (Lux Tool, Prague, Czech Republic) were used for fastening the plate and lid together with handmade wooden pads (size 8.5 × 13 × 2 mm) and the microtiter plates were incubated at 37 °C for 24 h. The MICs were evaluated by visual assessment of bacterial growth after colouring of metabolically active bacterial colonies with thiazolyl blue tetrazolium bromide dye (MTT) at a concentration of 600 μg/mL (Sigma-Aldrich, Prague, Czech Republic), when the interface of colour change from yellow to purple (relative to that of colours in control wells) was recorded in broth and agar. The MIC values were determined as the lowest concentrations that inhibited bacterial growth compared with the compound-free control and expressed in μg/mL (1024, 512, 256,128, 64, 32, 16, and 8 μg/mL, respectively). In the case of the vapour phase, considering a uniform distribution of the volatile compounds in the liquid and gaseous phase, these concentrations can be expressed as weight of volatile agent per volume unit of a well; therefore, MIC values would be 256, 128, 64, 32, 16, 8, 4, and 2 μg/cm^3^, respectively). The DMSO used as the negative control at a concentration of 1% did not inhibit any of the strains tested either in broth or agar media. All experiments were set in triplicates in three independent measurements, and results were expressed as median/modal MICs values. According to the widely accepted norm in MIC testing, the mode and median were used for the final value calculation when triplicate endpoints were within the two- and three-dilution ranges, respectively.

### 4.5. Chemical Analysis of EOs

For the characterisation of the EOs, GC/MS analysis was performed using the dual-column/dual-detector gas chromatograph Agilent GC-7890B system (Agilent Technologies, Santa Clara, CA, USA). equipped with autosampler Agilent 7693, two columns, a fused-silica HP-5MS column (30 m × 0.25 mm, film thickness 0.25 μm, Agilent 19091s-433) and a DB-HeavyWAX (30 m × 0.25 mm, film thickness 0.25 μm, Agilent 122–7132), and a flame ionisation detector (FID) coupled with single quadrupole mass selective detector Agilent MSD-5977B. The operational parameters were the following: helium as carrier gas at 1 mL/min, injector temperature 250 °C for both columns. The oven temperature was raised for both columns after 3 min from 50 to 280 °C. Initially, the heating velocity was 3 °C/min until the system reached a temperature of 120 °C. Subsequently, the velocity increased to 5 °C/min until a temperature of 250 °C, and after 5 min holding time, the heating speed reached 15 °C/min until obtaining a temperature of 280 °C. Heating was followed by an isothermic period of 20 min. The EO samples were diluted in n-hexane for GC/MS (Merck KGaA, Darmstadt, Germany) at the concentration 20 μL/mL. One microliter of the solution was injected in split mode in a split ratio 1:30. The mass detector was set to the following conditions: ionisation energy 70 eV, ion source temperature 230 °C, scan time 1 s, mass range 40–600 m/z.

The identification of constituents was based on comparison of their retention indices (RIs), retention times (RT), and spectra with the National Institute of Standards and Technology Library ver. 2.0.f and the available literature [57]. The RIs were calculated for compounds separated by the HP-5MS column using the retention times of n-alkanes series ranging from C8 to C40 (Sigma-Aldrich, Prague, Czech Republic). For each EO analysed, the final number of compounds was calculated as the sum of components simultaneously identified using both columns and the remaining constituents identified by individual columns only. Quantitative data are expressed as relative percentage content of constituents determined by the FID.

### 4.6. Chemical Analysis of EOs’ Vapour Phase

The analysis of the chemical composition of the headspace above a mixture of MH broth and *T. vulgaris* EO at a concentration of 512 μg/mL (i.e., the lowest MIC value obtained from the BMV assay) was performed using two different sampling techniques: HS-SPME and HS-GTS. Regardless of the sampling method used, for each experiment, a set of five samples were prepared, and a volume of 2 mL of the above-mentioned mixture was introduced into a 4 mL glass vial. Except for the first sample (t = 0 h), all EO samples were placed into an oven set at a temperature of 37 °C for incubation until their analysis at 3, 6, 9, and 12 h.

In HS-SPME, the headspace sampling was achieve using a fibre assembly coated with a 50/30 μm mixed layer of divinylbenzene/carboxen/polydimehylsiloxane (DVB/CAR/PDMS—SUPELCO, Bellefonte, PA, USA). When equilibrium was reached between the mixture and the headspace, the needle of the HS-SPME holder was inserted into the vial, and the coated fibre was exposed to the headspace for 15 min for adsorption of the volatile compounds. The needle was subsequently removed, inserted into the GC injector port, and set in splitless mode, where the desorption of analytes occurred. The injector temperature was set at 250 °C, and the fibre was left into the injector for the whole analysis until the next measure.

As for HS-GTS, however, the sampling technique was carried out using a 2.5 mL SampleLock gas tight syringe (Hamilton Bonaduz AG, Bonaduz, Switzerland), including a twist valve lock and a positive rear plunger stop to prevent sample loss. At equilibrium, with the valve of the syringe closed, the needle was passed through the vial septum and inserted until reaching the middle of the headspace. The valve was then opened, and a 2.5 mL sample was collected. Afterwards, the valve was closed again; the syringe was removed from the vial and inserted into the GC injector at a similar temperature of 250 °C but set in splitless mode. Finally, the valve was opened one more time to inject the headspace sample, and the syringe was immediately removed.

For both sampling methods, measurements were repeated every 3 h during a 12-h incubation period. Furthermore, analyses were performed on the HP-5MS column with similar operational parameters as described in Section 4.5 for GC/MS analyses, and their quantification was expressed as relative percentage content of constituents determined by the FID.

### 4.7. Statistical Analysis

The chemical analysis of the EO sample from supplier C (the most active EO based on the results of the antimicrobial assay) was performed in triplicate, including the chromatographic analysis of its liquid-phase and the headspace analysis using both extraction techniques (HS-SPME and HS-GTS). Relative peak area percentages were expressed as mean average of these three measurements ± standard deviation. For all EO samples’ chemical profiles to be compared with one another, chemical analysis of EO samples from suppliers A and B was carried out in one replication only.

## 5. Conclusions

To summarise, this study reports the antibacterial activity of three *T. vulgaris* EOs hydrodistilled from commercial samples of different origins against three standard bacterial strains associated with respiratory infections, namely, *H. influenzae*, *S. aureus*, and *S. pyogenes*, in both liquid and vapour phases when assayed using the BVM method. While all bacterial strains were sensitive to *T. vulgaris* EO vapours to a certain degree, Gram-negative strains of *H. influenzae* showed the highest susceptibility. The GC/MS analysis identified the EO samples as a thymol chemotype, whose major constituents were monoterpenoids such as thymol, carvacrol linalool, γ-terpinene, and *p*-cymene, but also sesquiterpenoids represented by β-caryophyllene. In opposition, the chemical analysis of the headspace reported fewer compounds in the vapour with a predominance of *p*-cymene, γ-terpinene, and α-pinene, whereas the amount of thymol was unusually low. As for which of the two headspace sampling techniques could prove more valuable for the chemical analysis of EO vapours, results show that both methods are rather complementary and interdependent: HS-SPME with optimised experimental conditions may yield more accurate results when aiming for qualitative aspects, while HS-GTS could provide more accurate data representing the true headspace distribution of the EO volatile agents, therefore proving to be a better technique when aiming for quantitative analysis. Further research, however, is needed to corroborate this argument. Overall, the potential of the procedures examined in this study could be further exploited to better assess the benefits of EO volatile compounds and their applications in the healthcare and pharmaceutical industries. Results of this study also suggest a potential of *T. vulgaris* EO for application in inhalation therapy against respiratory infections; however, a further pharmacological evaluation will be necessary in order to verify its potential practical use.

## Figures and Tables

**Figure 1 molecules-26-06553-f001:**
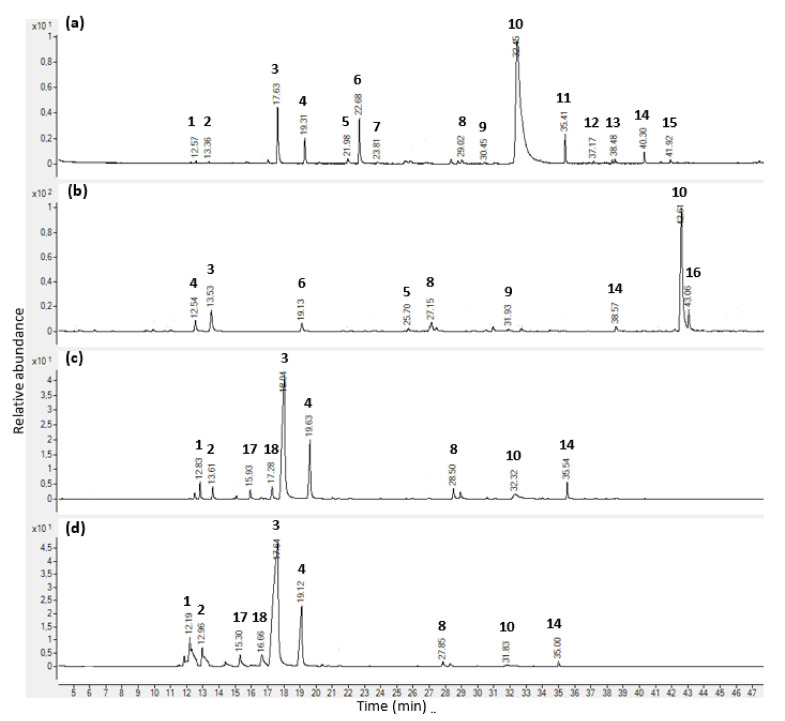
GC-MS chromatograms of *Thymus vulgaris* L. essential oil obtained from commercial source C and headspace analysis of its vapour. (**a**) Chromatogram on an HP-5MS column; (**b**) chromatogram on a DB-HeavyWAX column; (**c**) chromatogram of the headspace after a 9-h incubation period using solid-phase microextraction sampling technique; (**d**) chromatogram of the headspace after a 9-h incubation period using gas tight syringe sampling technique. Peak number and compound names: 1. α-pinene, 2. camphene, 3. *p*-cymene, 4. γ-terpinene, 5. linalool, 6. methyl octanoate, 7. Camphor, 8. Thymol methyl ether, 9. Carvone, 10. thymol. 11. β-caryophyllene, 12. γ-Muurolene, 13. δ-cadinene, 14. caryophyllene oxide, 15. α-epi-cadinol, 16. Carvacrol, 17. β-myrcene, 18. α-terpinene.

**Table 1 molecules-26-06553-t001:** In vitro growth-inhibitory effect of *Thymus vulgaris* L. essential oils in liquid and vapour phases against respiratory infection bacteria.

Supplier	Bacterium/Growth Medium/Minimum Inhibitory Concentration								
	*Staphylococcus aureus*			*Streptococcus pyogenes*			*Haemophilus influenzae*		
	Broth	Agar		Broth	Agar		Broth	Agar	
	(µg/mL)	(µg/mL)	(µg/cm^3^)	(µg/mL)	(µg/mL)	(µg/cm^3^)	(µg/mL)	(µg/mL)	(µg/cm^3^)
A	1024	1024	256	1024	1024	256	512	512	128
B	1024	1024	256	512	512	128	512	512	128
C	512	512	128	512	512	128	512	512	128
	Positive antibiotic control								
Oxacillin	0.25	>2	>0.5	NT	NT	NT	NT	NT	NT
Amoxicilin	NT	NT	NT	0.06	>2	>0.5	NT	NT	NT
Ampicilin	NT	NT	NT	NT	NT	NT	0.5	>16	>4

NT = Not tested.

**Table 2 molecules-26-06553-t002:** Chemical composition of *Thymus vulgaris* L. essential oils obtained from 3 different commercial suppliers.

RI ^a^		Compound	Cl. ^b^	Supplier/Column/Content (%)						Identification ^e^	
				A		B		C ^c^			
Obs.	Lit			HP-5MS	DB-WAX	HP-5MS	DB-WAX	HP-5MS	DB-WAX	HP-5MS	DB-WAX

761	783 ^d^	Methyl α-methylbutanoate	OA	0.08	- ^g^	0.08	-	0.05 ± 0.01	-	GC/MS	-
923	924	α-Thujene	MH	0.10	0.08	0.10	0.13	0.26 ± 0.02	0.24 ± 0.06	RI, GC/MS	GC/MS
929	939	α-Pinene	MH	0.62	0.38	0.61	0.62	0.43 ± 0.03	0.38 ± 0.07	RI, GC/MS	GC/MS
944	945	Camphene	MH	0.44	0.26	0.43	0.43	0.19 ± 0.01	0.18 ± 0.03	RI, GC/MS	GC/MS
972	969	Sabinene	MH	0.09	0.05	0.08	0.09	0.10 ± 0.01	0.09 ± 0.02	RI, GC/MS	GC/MS
977	979	1-octen-3-ol	OA	0.31	0.20	0.32	0.30	0.35 ± 0.02	0.35 ± 0.06	RI, GC/MS	GC/MS
989	988	β-Myrcene	MH	0.31	0.21	0.31	0.35	0.58 ± 0.04	0.58 ± 0.11	RI, GC/MS	GC/MS
994	996	3-Octanol	OA	0.05	-	tr.^f^	-	tr.	-	RI, GC/MS	-
1002	1002	α-Phellandrene	MH	0.06	-	0.06	0.05	0.10 ± 0.01	0.08 ± 0.02	RI, GC/MS	GC/MS
1007	1008	3-Carene	MH	-	-	-	-	tr.	-	RI, GC/MS	-
1014	1014	α-Terpinene	MH	0.62	-	0.61	-	1.07 ± 0.07	-	RI, GC/MS	-
1025	1020	*p*-Cymene	AH	22.15	16.43	22.03	25.35	12.75 ± 0.84	12.90 ± 2.55	RI, GC/MS	GC/MS
1027	1031	D-Limonene	MH	-	0.16	-	0.25	0.41 ± 0.03	0.23 ± 0.04	RI, GC/MS	GC/MS
1028	1023	*m*-Cymene	AH	0.46	-	0.47	-	-	0.18 ± 0.17	RI, GC/MS	GC/MS
1030	1032	1,8-Cineole	OM	0.69	0.51	0.70	0.76	0.39 ± 0.03	0.47 ± 0.10	RI, GC/MS	GC/MS
1058	1054	γ-Terpinene	MH	2.82	2.02	2.78	2.90	5.22 ± 0.33	5.40 ± 1.11	RI, GC/MS	GC/MS
1065	1068	*cis*-thujanol	OM	0.17	0.06	0.18	0.07	0.37 ± 0.03	0.14 ± 0.03	RI, GC/MS	GC/MS
1071	1078	*cis*-Linalool oxide	F	tr.	-	tr.	-	-	-	RI, GC/MS	-
1087	1086	Terpinolene	MH	0.18	-	0.17	-	0.11 ± 0.01	-	RI, GC/MS	-
1096	1102	*trans*-thujanol	OM	-	0.09	-	0.13	tr.	0.37 ± 0.08	RI, GC/MS	GC/MS
1100	1095	Linalool	OM	1.80	1.08	1.82	1.77	1.53 ± 0.07	1.38 ± 0.24	RI, GC/MS	GC/MS
1149	1141	Camphor	OM	0.32	0.23	0.18	0.34	0.06 ± 0.01	0.16 ± 0.03	RI, GC/MS	GC/MS
1165	1165	Endo-borneol	OM	1.14	-	1.15	-	0.63 ± 0.03	-	RI, GC/MS	-
1176	1174	Terpinen-4-ol	OM	0.61	-	0.61	-	0.62 ± 0.04	-	RI, GC/MS	-
1186	1180	m-Cymen-8-ol	OM	0.05	-	-	-	-	-	RI, GC/MS	-
1190	1183	*p*-Cymen-8-ol	OM	0.14	0.11	0.12	0.16	-	-	RI, GC/MS	GC/MS
1196	1189	α-Terpineol	OM	0.14	-	0.14	-	0.16 ± 0.02	-	RI, GC/MS	-
1204	1195	*cis*-Dihydrocarvone	OM	tr.	-	tr.	-	tr.	-	RI, GC/MS	-
1235	1235	Thymol methyl ether	OPM	1.15	2.11	1.15	3.01	0.76 ± 0.05	3.17 ± 0.67	RI, GC/MS	GC/MS
1244	1241	Carvacrol methyl ether	OPM	0.91	0.86	0.91	1.35	0.66 ± 0.04	1.09 ± 0.23	RI, GC/MS	GC/MS
1256	1242	Carvone	OM	1.39	1.05	1.40	1.38	1.01 ± 0.05	1.05 ± 0.24	RI, GC/MS	GC/MS
1269	1255	Geraniol	MH	0.05	-	0.05	0.09	0.05 ± 0.01	tr.	RI, GC/MS	GC/MS
1286	1282	Anethole	OPP	0.22	0.21	0.23	0.30	0.11 ± 0.06	0.07 ± 0.02	RI, GC/MS	GC/MS
1304	1290	Thymol	PM	38.42	29.52	38.93	41.31	48.09 ± 2.53	48.65 ± 10.80	RI, GC/MS	GC/MS
1310	1298	Carvacrol	PM	10.61	3.84	10.85	5.59	10.92 ± 3.50	3.60 ± 0.78	RI, GC/MS	GC/MS
1361	1357	Estragole	OPP	-	-	0.07	0.05	tr.	0.09 ± 0.02	RI, GC/MS	GC/MS
1379	1374	α-Copaene	SH	0.21	-	0.21	-	0.09 ± 0.01	-	RI, GC/MS	-
1388	1387	β-Bourbonene	SH	0.07	tr.	0.07	-	tr.	-	RI, GC/MS	GC/MS
1424	1418	β-Caryophyllene	SH	1.79	-	1.79	-	2.33 ± 0.13	-	RI, GC/MS	-
1433	1446 ^d^	Isogermacrene D	SH	-	-	-	-	tr.	-	GC/MS	-
1458	1452	α-Humulene	SH	0.09	tr.	0.08	0.07	0.11 ± 0.01	0.12 ± 0.02	RI, GC/MS	GC/MS
1475	1475	Geranyl propionate	OM	0.06	0.06	0.10	0.14	0.10 ± 0.01	0.11 ± 0.03	RI, GC/MS	GC/MS
1480	1478	γ-Muurolene	SH	0.19	0.15	0.21	0.22	0.21 ± 0.02	0.21 ± 0.04	RI, GC/MS	GC/MS
1483	1493	α-Amorphene	SH	-	-	tr.	-	tr.	-	RI, GC/MS	-
1498	1495	Valencene	SH	0.09	-	0.09	-	0.11 ± 0.01	-	RI, GC/MS	-
1503	1499	α-Muurolene	SH	0.10	-	0.10	-	0.09 ± 0.01	-	RI, GC/MS	-
1510	1509	β-Bisabolene	SH	0.06	-	0.06	-	0.06 ± 0.01	-	RI, GC/MS	-
1518	1513	γ-Cadinene	SH	0.38	0.43	0.39	0.58	0.33 ± 0.02	0.73 ± 0.16	RI, GC/MS	GC/MS
1527	1524	δ-Cadinene	SH	0.48	-	0.48	-	0.50 ± 0.03	-	RI, GC/MS	-
1561	1564	Nerolidol	OS	tr.	-	-	-	tr.	-	RI, GC/MS	-
1585	1578	Spathulenol	OS	-	-	tr.	-	tr.	-	RI, GC/MS	-
1591	1581	Caryophyllene oxide	OS	1.31	0.92	1.32	1.24	0.68 ± 0.03	0.66 ± 0.15	RI, GC/MS	GC/MS
1617	1606	Humulene epoxide II	OS	0.06	tr.	0.06	0.05	tr.	-	RI, GC/MS	GC/MS
1621	1627	Epicubenol	OS	0.06	tr.	0.06	-	tr.	tr.	RI, GC/MS	GC/MS
1628	1630	γ-Eudesmol	OS	0.13	0.10	0.13	0.13	0.13 ± 0.01	0.13 ± 0.03	RI, GC/MS	GC/MS
1634	1642	Cubenol	OS	tr.	-	-	0.09	tr.	-	RI, GC/MS	GC/MS
1648	1640	α-epi-Cadinol	OS	0.48	tr.	0.49	-	0.36 ± 0.03	0.05 ± 0.05	RI, GC/MS	GC/MS
1652	1645	δ-Cadinol	OS	-	0.06	-	-	tr.	tr.	RI, GC/MS	GC/MS
1661	1653	α-Cadinol	OS	0.05	-	0.05	-	0.07 ± 0.02	-	RI, GC/MS	-
1682	1677	Cadalene	SH	0.12	-	0.14	-	tr.	-	RI, GC/MS	-
1844	1844	Perhydrofarnesyl acetone	OM	tr.	-	tr.	-	tr.	-	RI, GC/MS	-
^h^	1001	4-Carene	MH	-	-	-	0.64	-	1.05 ± 0.23	-	GC/MS
^h^	NA	4-Pentenyl butyrate	OM	-	tr.	-	0.06	-	-	-	GC/MS
^h^	1351	α-Cubebene	SH	-	0.06	-	0.10	-	0.07 ± 0.01	-	GC/MS
^h^	1290	Isobornyl acetate	OM	-	0.08	-	0.11	-	-	-	GC/MS
^h^	1521	Calamenene	SH	-	-	-	0.19	-	tr.	-	GC/MS
^h^	1156	Isoborneol	OM	-	0.86	-	1.17	-	0.74 ± 0.16	-	GC/MS
^h^	1455	Aromandendrene	SH	-	tr.	-	0.05	-	tr.	-	GC/MS
^h^	NA	Lavandulyl butyrate	OM	-	0.05	-	0.07	-	tr.	-	GC/MS
^h^	1372	*p*-Cymen-7-ol	OM	-	tr.	-	0.05	-	-	-	GC/MS
^h^	2105	Phytol	OD	-	0.07	-	0.23	-	tr.	-	GC/MS
		Total identified (%)		99.55	99.30	99.65	99.68	99.62	99.61		

^a^ RI = retention indices. Obs. = retention indices determined relative to a homologous series of *n*-alkanes (C8–C40) on an HP-5MS column. Lit. = literature RI values [49,50]; ^b^ Cl = class; AH—aromatic hydrocarbon, F—furanoid, MH—monoterpene hydrocarbon, OA—oxygenated aliphatic, OD—oxygenated diterpene, OM—oxygenated monoterpene, OPM—oxygenated phenolic monoterpene, OPP—oxygenated phenylpropanoid, PM—phenolic monoterpene, OS—oxygenated sesquiterpene, SH—sesquiterpene hydrocarbon. ^c^ Relative peak area percentage as mean of three measurements ± standard deviation. ^d^ Literature RI values [51]. ^e^ Identification method: GC/MS = mass spectrum was identical to that of National Institute of Standards and Technology Library (ver. 2.0.f); RI = the retention index was matching literature database. ^f^ tr. = traces, relative peak area < 0.05%. ^g^ - = not detected. ^h^ Retention indices were not calculated for compounds determined by DB-WAX column.

**Table 3 molecules-26-06553-t003:** Chemical composition of the headspace above a mixture of Mueller–Hinton broth and *Thymus vulgaris* L. essential oils at a concentration of 512 µg/mL over a 12-h period using solid-phase microextraction sampling technique.

RI ^a^		Compounds	Cl. ^b^	Supplier/Time (h)/Content (%)														
				A					B					C ^c^				
Obs.	Lit.			0	3	6	9	12	0	3	6	9	12	0	3	6	9	12
921	921	Tricyclene	MH	0.05	0.05	0.05	0.05	0.05	tr. ^e^	tr.	0.05	0.07	0.07	0.08 ± 0.02	0.08 ± 0.01	0.07 ± 0.02	0.10 ± 0.01	0.08 ± 0.02
927	924	α-Thujene	MH	0.95	0.91	0.83	0.84	0.93	0.97	0.79	0.98	1.10	1.07	1.01 ± 0.05	1.00 ± 0.03	0.93 ± 0.03	0.94 ± 0.13	0.94 ± 0.05
933	939	α-Pinene	MH	2.37	2.25	2.18	2.11	2.37	1.97	1.60	2.23	2.45	2.41	2.73 ± 0.18	2.69 ± 0.13	2.57 ± 0.26	2.82 ± 0.30	2.70 ± 0.26
949	945	Camphene	MH	1.47	1.42	1.37	1.33	1.51	0.85	0.83	1.58	1.69	1.67	1.58 ± 0.56	1.56 ± 0.53	1.52 ± 0.58	2.13 ± 0.16	1.82 ± 0.57
976	969	Sabinene	MH	tr.	tr.	- ^f^	tr.	tr.	tr.	tr.	0.05	0.06	0.06	tr.	tr.	tr.	0.07 ± 0.00	0.06 ± 0.02
978	974	β-Pinene	MH	0.49	0.47	0.42	0.45	0.48	0.43	0.40	0.49	0.51	0.53	0.45 ± 0.01	0.49 ± 0.06	0.48 ± 0.07	0.56 ± 0.04	0.51 ± 0.06
991	988	3-Octanone	OA	tr.	tr.	-	tr.	tr.	-	-	tr.	tr.	tr.	-	-	-	-	-
995	988	β-Myrcene	MH	2.76	2.70	2.63	2.48	2.72	1.53	1.45	1.79	1.88	1.88	1.62 ± 0.32	1.53 ± 0.18	1.51 ± 0.29	1.83 ± 0.03	1.62 ± 0.26
1009	1002	α-Phellandrene	MH	0.36	0.35	0.34	0.33	0.35	0.19	0.19	0.20	0.20	0.20	0.26 ± 0.03	0.26 ± 0.02	0.25 ± 0.03	0.28 ± 0.01	0.24 ± 0.03
1013	1008	3-Carene	MH	0.21	0.21	0.21	0.20	0.22	0.09	0.08	0.13	0.14	0.14	0.11 ± 0.04	0.10 ± 0.03	0.10 ± 0.03	0.14 ± 0.01	0.12 ± 0.03
1021	1014	α-Terpinene	MH	3.76	3.72	3.61	3.58	3.72	1.61	1.58	2.34	2.37	2.38	2.20 ± 0.66	2.17 ± 0.64	2.14 ± 0.65	2.92 ± 0.05	2.46 ± 0.64
1036	1020	*p*-Cymene	AH	58.61	56.24	55.41	54.57	56.35	74.22	74.50	71.03	71.16	69.91	67.21 ± 7.03	66.38 ± 7.46	65.89 ± 6.38	58.38 ± 0.23	60.62 ± 6.81
1055	1023	*m*-Cymene	AH	tr.	0.05	0.05	tr.	tr.	-	-	-	-	-	-	-	-	tr.	tr.
1067	1054	γ-Terpinene	MH	19.7	20.45	20.30	20.23	20.18	7.07	7.23	9.30	9.18	9.17	12.37 ± 2.86	12.51 ± 2.64	12.67 ± 3.02	16.18 ± 0.16	14.14 ± 2.88
1094	1086	Terpinolene	MH	0.31	0.33	0.31	0.32	0.32	0.10	0.11	0.24	0.24	0.24	0.16 ± 0.10	0.16 ± 0.10	0.17 ± 0.10	0.27 ± 0.01	0.21 ± 0.09
1101	1089	*p*-Cymenene	MH	0.20	0.25	0.19	0.21	0.22	0.15	0.14	0.19	0.21	0.20	0.12 ± 0.06	0.10 ± 0.04	0.09 ± 0.02	0.14 ± 0.01	0.11 ± 0.03
1115	1095	Linalool	OM	0.27	0.26	0.25	0.26	0.33	0.13	0.10	0.28	0.34	0.34	0.23 ± 0.21	0.16 ± 0.12	0.16 ± 0.13	0.29 ± 0.03	0.19 ± 0.10
1153	1141	Camphor	OM	0.05	tr.	tr.	0.05	0.05	tr.	tr.	0.06	0.06	0.06	tr.	tr.	0.06 ± 0.04	0.09 ± 0.01	0.08 ± 0.04
1186	1165	Endo-borneol	OM	tr.	tr.	tr.	tr.	tr.	0.08	0.07	tr.	0.05	0.05	0.11 ± 0.02	0.09 ± 0.01	0.08 ± 0.01	0.07 ± 0.01	0.08 ± 0.01
1192	1174	Terpinen-4-ol	OM	0.06	0.08	0.08	0.10	0.10	tr.	tr.	0.08	0.08	0.09	tr.	tr.	0.06 ± 0.05	0.10 ± 0.01	0.09 ± 0.04
1213	1195	Estragole	OPP	tr.	tr.	tr.	tr.	tr.	-	-	-	-	tr.	0.10 ± 0.06	0.13 ± 0.09	0.11 ± 0.04	0.17 ± 0.03	0.18 ± 0.06
1247	1235	Thymol methyl ether	OPM	1.78	2.48	2.68	2.82	2.44	3.05	3.77	2.68	2.24	2.58	1.83 ± 0.13	2.41 ± 0.17	2.77 ± 0.41	2.61 ± 0.17	2.91 ± 0.34
1257	1241	Carvacrol methyl ether	OPM	1.15	1.52	1.68	1.68	1.51	1.67	2.02	1.55	1.40	1.49	1.08 ± 0.12	1.37 ± 0.11	1.50 ± 0.14	1.54 ± 0.03	1.69 ± 0.11
1293	1287	Bornyl acetate	OM	0.17	0.24	0.29	0.33	0.21	0.05	0.08	0.15	0.11	0.14	tr.	0.15 ± 0.08	0.15 ± 0.06	0.25 ± 0.04	0.24 ± 0.11
1306	1284	Anethol	OPP	tr.	tr.	tr.	0.05	tr.	-	-	-	-	-	tr.	tr.	0.08 ± 0.00	0.12 ± 0.02	0.12 ± 0.03
1339	1290	Thymol	PM	2.61	2.80	3.14	3.36	3.12	5.25	3.50	2.13	2.64	3.15	5.27 ± 1.27	4.57 ± 0.81	4.62 ± 0.84	3.72 ± 0.07	5.05 ± 0.45
1368	1298	Carvacrol	PM	tr.	tr.	tr.	tr.	tr.	-	-	-	-	-	-	tr.	tr.	tr.	tr.
1381	1372	*p*-Cymen-7-ol	OM	tr.	tr.	tr.	tr.	tr.	-	-	tr.	tr.	tr.	-	-	-	tr.	tr.
1386	1374	α-Copaene	SH	0.09	0.13	0.17	0.19	0.12	tr.	tr.	0.11	0.07	0.11	tr.	tr.	0.07 ± 0.08	0.16 ± 0.01	0.19 ± 0.05
1395	1387	β-Bourbonene	SH	tr.	tr.	0.05	0.07	tr.	tr.	tr.	tr.	tr.	0.05	tr.	tr.	tr.	0.09 ± 0.01	0.10 ± 0.01
1434	1418	β-Caryophyllene	SH	1.59	2.14	2.74	3.09	1.94	0.26	0.32	1.59	1.13	1.49	0.77 ± 0.82	1.19 ± 1.26	1.28 ± 1.29	2.94 ± 0.32	2.40 ± 1.59
1442	1446 ^d^	isogermacrene D	SH	tr.	0.06	0.09	tr.	tr.	-	-	tr.	tr.	tr.	-	tr.	tr.	0.08 ± 0.06	tr.
1469	1452	α-Humulene	SH	tr.	tr.	0.07	0.07	tr.	-	tr.	tr.	tr.	tr.	tr.	tr.	tr.	0.07 ± 0.01	0.06 ± 0.03
1475	1465	*cis*-muurola-4(14),5-diene	SH	-	tr.	tr.	tr.	-	-	-	tr.	-	tr.	-	-	-	tr.	tr.
1482	1475	Geranyl propionate	OM	tr.	tr.	tr.	tr.	-	-	-	-	-	-	-	-	-	-	tr.
1491	1478	γ-Muurolene	SH	0.06	0.08	0.12	0.14	0.07	tr.	tr.	0.06	tr.	0.06	tr.	0.07 ± 0.04	0.08 ± 0.05	0.12 ± 0.01	0.11 ± 0.05
1509	1480	Germacrene D	SH	tr.	tr.	tr.	tr.	-	-	-	tr.	-	-	-	-	-	tr.	tr.
1512	1491	Valencene	SH	-	tr.	tr.	tr.	-	-	-	-	-	-	-	-	-	tr.	tr.
1515	1499	α-Muurolene	SH	tr.	tr.	tr.	tr.	tr.	-	-	tr.	tr.	tr.	-	-	-	tr.	tr.
1520	1509	β-Bisabolene	SH	tr.	tr.	tr.	tr.	tr.	-	-	tr.	-	tr.	-	-	-	tr.	tr.
1532	1513	γ-Cadinene	SH	0.08	0.08	0.12	0.14	0.07	tr.	tr.	0.07	tr.	0.07	tr.	0.10 ± 0.04	0.08 ± 0.03	0.12 ± 0.02	0.12 ± 0.05
1538	1524	δ-Cadinene	SH	0.1	0.1	0.15	0.17	0.09	tr.	tr.	0.05	tr.	0.05	tr.	0.07 ± 0.04	0.08 ± 0.04	0.12 ± 0.02	0.11 ± 0.05
1541	1521	Calamenene	SH	tr.	tr.	-	0.07	tr.	tr.	0.05	0.05	0.03	0.05	tr.	0.07 ± 0.01	0.06 ± 0.01	0.07 ± 0.01	0.07 ± 0.01
1605	1581	Caryophyllene oxide	OS	tr.	tr.	tr.	tr.	tr.	-	-	-	tr.	tr.	-	-	tr.	tr.	tr.
		Total identified (%)		99.68	99.83	99.91	99.77	99.85	99.89	99.18	99.72	99.64	99.89	99.33	99.87	99.87	99.74	99.63

^a^ RI = retention indices. Obs. = retention indices determined relative to a homologous series of n-alkanes (C8-C40) on an HP-5MS column. Lit. = literature Ri values [49,50]; ^b^ Cl = class; AH—aromatic hydrocarbon, MH—monoterpene hydrocarbon, OA—oxygenated aliphatic, OM—oxygenated monoterpene, OPM—oxygenated phenolic monoterpene, OPP—oxygenated phenylpropanoid, PM—phenolic monoterpene, OS—oxygenated sesquiterpene, SH—sesquiterpene hydrocarbon. ^c^ Relative peak area percentage as mean of three measurements ± standard deviation. ^d^ Literature RI values from [51]. ^e^ tr. = traces, relative peak area < 0.05%. ^f^ - = not detected.

**Table 4 molecules-26-06553-t004:** Chemical composition of the headspace above a mixture of Mueller–Hinton broth and *Thymus vulgaris* L. essential oils at a concentration of 512 µg/mL over a 12-h period using gas tight syringe headspace sampling technique.

RI ^a^		Compounds	Cl. ^b^	Supplier/Time (h)/Content (%)														
				A					B					C ^c^				
Obs.	Lit.			0	3	6	9	12	0	3	6	9	12	0	3	6	9	12
-	NA	2-Ethyl furan	F	tr. ^d^	tr.	tr.	-^e^	tr.	0.09	0.09	0.09	0.07	0.04	0.09 ± 0.05	0.07 ± 0.01	0.08 ± 0.04	0.08 ± 0.02	0.07 ± 0.00
778	NA	Methyl α-methylbutyrate	OA	0.08	0.10	0.13	0.08	0.12	0.22	0.17	0.23	0.16	0.11	0.10 ± 0.07	0.08 ± 0.02	0.10 ± 0.04	0.07 ± 0.01	0.09 ± 0.00
912	921	Tricyclene	MH	0.11	0.20	0.16	0.12	0.19	0.17	0.15	0.06	0.20	0.18	0.19 ± 0.01	0.17 ± 0.02	0.19 ± 0.01	0.20 ± 0.04	0.24 ± 0.00
918	924	α-Thujene	MH	2.01	2.17	2.03	1.75	2.23	2.09	2.03	2.11	2.38	2.34	1.97 ± 0.24	1.89 ± 0.07	1.95 ± 0.15	1.91 ± 0.17	2.30 ± 0.00
924	939	α-Pinene	MH	6.19	5.70	6.83	4.63	5.74	8.98	7.75	7.84	6.79	6.34	7.96 ± 2.05	6.94 ± 1.11	6.90 ± 1.73	7.20 ± 0.33	7.11 ± 0.00
939	945	Camphene	MH	3.40	3.21	3.57	2.69	3.28	4.59	4.04	4.07	3.81	3.71	4.59 ± 0.91	4.22 ± 0.35	4.16 ± 0.76	4.24 ± 0.03	4.45 ± 0.00
953	969	Sabinene	MH	-	-	-	0.07	-	-	-	-	-	-	tr.	tr.	0.05 ± 0.02	0.13 ± 0.19	0.06 ± 0.00
967	980	β-Pinene	MH	0.91	0.83	0.85	0.78	0.87	1.07	0.88	0.99	0.89	0.81	0.75 ± 0.37	0.94 ± 0.04	0.91 ± 0.11	0.74 ± 0.25	0.81 ± 0.00
984	988	β-Myrcene	MH	2.88	2.80	3.14	3.13	2.88	1.88	1.89	1.98	1.90	2.06	1.82 ± 0.06	1.96 ± 0.24	1.88 ± 0.08	1.63 ± 0.15	2.09 ± 0.00
998	1002	α-Phellandrene	MH	0.31	0.35	0.23	0.20	0.35	0.05	0.11	0.11	0.10	0.07	0.13 ± 0.05	0.23 ± 0.06	0.20 ± 0.10	0.19 ± 0.03	0.22 ± 0.00
1002	1008	3-Carene	MH	0.28	0.25	0.36	0.13	0.26	0.30	0.12	0.11	0.08	0.05	0.14 ± 0.10	0.10 ± 0.05	0.13 ± 0.08	0.13 ± 0.05	0.16 ± 0.00
1010	1014	α-Terpinene	MH	3.64	3.57	2.57	4.17	3.80	2.21	2.06	1.93	2.12	2.34	2.57 ± 0.18	2.87 ± 0.50	2.77 ± 0.33	2.41 ± 0.18	2.88 ± 0.00
1028	1020	*p*-Cymene	AH	45.93	47.05	47.48	50.73	48.78	52.28	54.29	56.81	57.30	57.41	44.80 ± 1.03	49.19 ± 2.67	49.20 ± 3.00	46.10 ± 2.10	49.28 ± 0.00
1044	1023	*m*-Cymene	AH	tr.	tr.	0.27	0.28	tr.	-	-	-	-	-	0.07 ± 0.07	0.09 ± 0.06	tr.	0.06 ± 0.08	tr.
1058	1054	γ-Terpinene	MH	13.40	13.90	15.46	17.54	14.80	5.76	6.22	6.58	6.69	6.99	9.54 ± 1.00	11.77 ± 1.52	11.82 ± 1.70	10.32 ± 0.68	11.85 ± 0.00
1083	1086	Terpinolene	MH	0.18	0.19	0.23	0.26	0.20	0.10	0.12	0.14	0.14	0.14	0.13 ± 0.02	0.18 ± 0.04	0.18 ± 0.03	0.14 ± 0.02	0.17 ± 0.00
1089	1089	*p*-Cymenene	MH	0.10	0.09	0.13	0.14	0.09	0.08	0.09	0.10	0.09	0.09	0.06 ± 0.00	0.08 ± 0.01	0.07 ± 0.01	0.06 ± 0.02	0.06 ± 0.00
1103	1095	Linalool	OM	0.09	0.08	0.15	0.16	0.10	0.08	0.06	0.12	0.11	0.10	0.09 ± 0.03	0.09 ± 0.03	0.09 ± 0.01	0.06 ± 0.03	0.08 ± 0.00
1133	1141	Camphor	OM	tr.	tr.	tr.	tr.	tr.	tr.	tr.	tr.	tr.	tr.	tr.	tr.	tr.	tr.	tr.
1171	NA	2-Ethyl-5-methylfuran	F	tr.	tr.	tr.	tr.	tr.	tr.	-	tr.	-	tr.	tr.	tr.	tr.	tr.	tr.
1180	1174	Terpinen-4-ol	OM	tr.	tr.	tr.	tr.	tr.	tr.	tr.	tr.	tr.	tr.	tr.	tr.	tr.	tr.	tr.
1200	1195	Estragole	OPP	-	-	tr.	tr.	-	-	-	-	-	-	tr.	tr.	tr.	tr.	tr.
1234	1235	Thymol methyl ether	OPM	0.34	0.34	0.62	0.67	0.43	0.24	0.39	0.45	0.43	0.49	0.31 ± 0.12	0.53 ± 0.15	0.57 ± 0.12	0.43 ± 0.08	0.50 ± 0.14
1244	1241	Carvacrol methyl ether	OPM	0.21	0.19	0.38	0.40	0.25	0.13	0.23	0.25	0.25	0.28	0.19 ± 0.08	0.30 ± 0.09	0.33 ± 0.08	0.24 ± 0.05	0.29 ± 0.08
1281	1287	Bornyl acetate	OM	tr.	tr.	0.05	0.05	tr.	-	tr.	tr.	tr.	tr.	tr.	tr.	tr.	tr.	tr.
1328	1290	Thymol	PM	0.50	0.27	0.58	0.60	0.28	0.25	0.10	0.28	0.26	0.22	0.46 ± 0.31	0.51 ± 0.36	0.55 ± 0.30	0.25 ± 0.10	0.58 ± 0.37
1372	1374	α-Copaene	SH	tr.	tr.	tr.	tr.	tr.	-	tr.	tr.	-	tr.	tr.	tr.	tr.	tr.	tr.
1381	1387	β-Bourbonene	SH	-	-	tr.	tr.	-	-	-	-	-	-	-	tr.	tr.	-	tr.
1417	1418	β-Caryophyllene	SH	0.17	0.17	0.38	0.34	0.19	0.07	0.11	0.14	0.13	0.15	0.21 ± 0.11	0.37 ± 0.15	0.41 ± 0.00	0.28 ± 0.08	0.39 ± 0.20
1475	1478	γ-Muurolene	SH	-	-	tr.	tr.	-	-	-	-	-	tr.	tr.	tr.	tr.	tr.	tr.
1513	1513	γ-Cadinene	SH	-	-	tr.	tr.	-	-	-	-	-	-	-	tr.	tr.	-	tr.
1521	1524	δ-Cadinene	SH	-	-	tr.	tr.	-	-	-	-	-	-	-	tr.	tr.	-	tr.
		Total identified (%)		99.95	99.80	99.94	99.94	99.95	99.98	99.86	99.90	99.92	99.71	99.87	99.83	99.90	99.90	99.87

^a^ RI = retention indices. Obs. = retention indices determined relative to a homologous series of *n*-alkanes (C8–C40) on an HP-5MS column. Lit. = literature Ri values [49,50,51]. ^b^ Cl = Class; AH—aromatic hydrocarbon, MH—monoterpene hydrocarbon, OA—oxygenated aliphatic, OM—oxygenated monoterpene, OPM—oxygenated phenolic monoterpene, OPP—oxygenated phenylpropanoid, PM—phenolic monoterpene, OS—oxygenated sesquiterpene, SH—sesquiterpene hydrocarbon. ^c^ Relative peak area percentage as mean of three measurements ± standard deviation. ^d^ tr. = traces, relative peak area < 0.05%. ^e^ - = not detected.

## Data Availability

Not applicable.

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
