# Peer review of "Antibacterial Activity of Thymus vulgaris L. Essential Oil Vapours and Their GC/MS Analysis Using Solid-Phase Microextraction and Syringe Headspace Sampling Techniques"

_molecules, 2021, doi:10.3390/molecules26216553_

Round 1
Reviewer 1 Report
Natural essential oils have served people for thousands of years. Already in antiquity, oils were distilled and used both for body care and healing treatments. Today, little has changed in this regard.
The authors of the manuscript examined Thymus vulgaris Essential Oil. The work is generally well written. Introduction, methodology, research results, discussion and conclusions are correctly written. However, the work would gain value if the authors included a supplement with examples of GC / MS chromatograms.
Author Response
Dear Reviewer,
Thank you for your comments,
Please see the attachment.
Kind Regards,
Julien Antih

Reviewer 2 Report
Everywhere in the text:
- the words “in vitro” and the names of the plants should be written in italic.
- the names of the plants should be written correctly: the names of the authors are missing.
In: Abstract
- some corrections of the text are necessary, because it is not clear enough now.
In: Introduction
- the aim of the paper is not clear.
In: Results and Discussion
- the information about the yield, color, and odor of the essential oils is missing.
- it should be given the explanation about the difference of the essential oil yields and the concentration of volatile compounds.
- the explanation about different anti-bacterial activities of the samples is missing. The discussion of the data should be generalized on a microorganism group basis - Gram positive and Gram negative bacteria.
- in the text and in the tables the classification of the compounds is incorrect, for example: p-cymene is an aromatic hydrocarbons; thymol and carvacrol are phenols; 3-octanol is an aliphatic alcohol; eugenol is a phenylpropanoid and a member of phenols; caryophyllene oxide is an oxygenated sesquiterpenes; phytol is an acyclic diterpene alcohols, etc. (Bauer K., D. Garbe, H. Surburg – Common fragrance and flavor materials. Preparation, properties and uses, fourth completely revised Edition, Weinheim, New York, Chichester, Brisbane, Singapore, Toronto, Wiley – VCH, 2001).
In: Material and methods
- the size of the plants should be specified.
In: Conclusions
- some corrections of the text are necessary, because it is not clear enough now.
in: References
- the list of references should be written according to the guideline for authors.
Author Response
Dear Reviewer,
Thank you for your comments.
Please see the attachment.
Kind regards,
Julien Antih

Round 2
Reviewer 2 Report
some corrections of the classification of the compounds are necessary, for example:
- oxygenated aliphatics
- monoterpene hyrocarbons
- oxygenated monoterpenes
- sesquiterpene hydrocarbons
- oxygenated sesquiterpenes
- phenyl propanoid hydrocrbons
- oxygenated phenyl propanoids
- oxygenated diterpenes
Group of oxygenated components are ketones, ethers, alcohols, phenols, esters, and oxides.
Author Response
Dear Reviewer,
Please see the attachment.
Kind Regards,
Julien Antih
